# Experiences of Homeless Families in Parenthood: A Systematic Review and Synthesis of Qualitative Evidence

**DOI:** 10.3390/ijerph17082712

**Published:** 2020-04-15

**Authors:** Filipa Maria Reinhardt Andrade, Amélia Simões Figueiredo, Manuel Luís Capelas, Zaida Charepe, Sérgio Deodato

**Affiliations:** 1 Institute of Health Sciences, Doctorate Student of the Doctorate Degree in Nursing, Universidade Católica Portuguesa, 1649-023 Lisbon, Portugal; 2Institute of Health Sciences, Centre for Interdisciplinary Research in Health (CIIS), Universidade Católica Portuguesa, 1649-023 Lisbon, Portugal; simoesfigueiredo@ics.lisboa.ucp.pt (A.S.F.); luis.capelas@ics.lisboa.ucp.pt (M.L.C.); Zaidacharepe@ics.lisboa.ucp.pt (Z.C.); sdeodato@ics.lisboa.ucp.pt (S.D.)

**Keywords:** family, homeless, nursing, parenting, parents, vulnerable population

## Abstract

The objective of this systematic review was to identify the available qualitative data and to develop a framework to address the life experiences of homeless families in parenthood. The research was performed in the PubMed and CINAHL Complete databases, for works published in Portuguese, English, French and Spanish. Studies that included qualitative data, or both qualitative and quantitative data, were considered for this research. A total of 358 articles were obtained, of which 37 were assessed for eligibility, and 26 were rejected. In the end, 11 studies were selected. The Joanna Briggs Institute Critical Appraisal Checklist for Qualitative Research was used. These studies were conducted mostly in the United States, in temporary/transitional shelters for nuclear or single-parent families (led by women) in a homeless situation. In this context, the area which arose as the more relevant one was mental health, followed by the social studies. Two types of dimensions emerged from the results: mediating dimensions (which include the categories “Insecurity”, “Lack of Privacy”, “Isolation”, “Stigma” and “Disempowerment”) that are responsible for difficulties related to education, and behavioural changes in both the parents and the children; and supporting dimensions (which include the categories “Context as a Facilitator”, “Relationship with Others” and “Parents’ and children’s Self”) that lead to motivation, as well as the acquisition of strategies by the parents, to resolve parenting issues. This research helps expand nursing knowledge and presents a synthesis of the life experiences of homeless families in parenthood. Nursing can respond to the vulnerable population, due to its predominant role in promoting their health.

## 1. Introduction

Parenthood is considered “the recognition of a mother’s/father’s responsibilities, the assumption of behaviours that promote the children’s growth and development, as well as the internalisation of expectations expressed by individuals, families, friends and society, regarding the parental role’s appropriate or inappropriate behaviours”(own translation, from: Ordem dos Enfermeiros) [1] (p. 66). Parenthood is understood as a transition [2,3,4]. Life transitions correspond to periods of greater vulnerability and increased health risk, which is why they have been increasingly considered a central concept for nursing [2,5]. Nurses are aware of the needs and changes brought upon by transitions, and they prepare the patient/family to better cope with such events, through learning and acquiring new skills [2].

The transition experience requires, as outcome indicators: the incorporation of new knowledge; the change of behaviours; the redefinition of the meanings associated with the occurred events; and, consequently, the redefinition of oneself within the social context [5]. Concerning possible constraints to the transition process, the author refers that, although community resources are able to create conditions which facilitate transition (family support, relevant information, referral, decision-making support), they may also create transition-inhibiting conditions (inappropriate counselling, insufficient information, etc.). The author points out that the society’s development stage has a similar dual effect on transition—it may either facilitate the process, through the creation of laws and regulations, or inhibit it, through stigma, stereotypes, and marginalization [3].

In this regard, parenthood is considered to be an important, complex, challenging and highly responsible task for the human being, since it concerns the preparation of another human being for the challenges of development, namely physical, economical and psycho-social adversities [6]. While being an activity that usually involves the children, the parents and other family members in an interaction throughout life, it is not exclusive for these players and may involve other people, such as nurses, teachers, friends, partners, and even strangers [7]. In this sense, any person that participates in the care and development of a child is part of the parenthood process [8]. This concept emphasises that, in order to acquire the sense of being a father, or a mother, and to meet all the relevant requirements, it is not enough to just become a parent. It is necessary to go through a complex, conscious—and sometimes unconscious—process of role appropriation. Although parenthood is a subjective concept, because it is influenced by cultural beliefs, the dimensions and structural tasks of the parent-child relationship tend to remain similar, thus allowing its assessment (i.e., the evaluation of parental skills and competences) and producing data adequate for research purposes [6]. In this sense, Cruz considers five assessable parental functions: (1) Satisfaction of the basic survival and health needs—Where he emphasizes that the inability to assume this function may be related to socio-economic constraints, natural disasters, drug addiction, or parents’ disease; (2) Providing of an organized and predictable physical world for the child—Which includes safety-promoting routines (severely dysfunctional families reveal a great difficulty in performing this function); (3) Responding to the needs of cognitive understanding with respect to extra-family realities—Stresses the difficulty of families that are more isolated or less permeable in ensuring this function; (4) Satisfaction of the needs of affection, trust and safety—This function is associated with the construction of attachment, which has proved to be a fundamental predictor of personal adaptation throughout life; (5) Satisfaction of the child’s social interaction needs—The family should perform this function because, as the original nucleus of the child’s socialization, it influences the future integration of the child in broader social contexts [9].

Furthermore, Figueiredo uses the life cycle stages (“Family with young children”, “Family with children at school”, and “Family with adolescent children”) to define several assessment categories with respect to the parental role’s competence. The aforesaid categories differ in terms of knowledge dimension, as well as adhesion behaviour. This author relates the knowledge of the role with the available information and the ability to learn skills, in order to identify the need to acquire new knowledge conducive to behavioural changes. She also relates the adhesion behaviour dimension to those aspects which highlight the parents’ actions that reflect the full incorporation of parenthood through the identity of the role. That incorporation is evidenced by trust and competence in the role’s performance. The evaluative categories of knowledge and adhesion behaviour are related to: physical care (which includes nourishment, elimination, personal hygiene and physical exercise); leisure activities; health surveillance; safety and prevention of accidents; attachment; cognitive, psychosexual, emotional and social development; structuring rules; a positive interaction with the child; adaptation to school; and the parents’ involvement in education [10].

When we observe the different conditions that should be involved in the parental process/activity, the complexity associated with the process of educating a child becomes evident. In this sense, family plays a key role: it constitutes the most immediate space for the child’s care, affections, dependence and socialization; it defines habits, health care routines, education, and the contexts to which the child is exposed (friends, school, neighbourhood, etc.); and it influences the present, the future and the development of the child. However, the family is not always able to provide a homogeneous environment and can have a negative impact on the child’s well-being [11].

Within the family, there may be risk factors (internal or external), which represent a deficit in the children’s development potential. On the other hand, the existence of protective factors (internal or external) that encourage the full development of the child can also occur. The internal protection factors relate to the quality of the interactions between family members. or to the involvement in stimulant activities. The external protective factors are associated with socioeconomic patterns, such as the family’s structure, the parents’ adequate working conditions, or the social support provided by public policies. As for external risk factors, they are related to demographic variables (e.g., having a single/adolescent mother, father’s absence, separations and divorces) and to socioeconomic patterns (e.g., poverty, inequalities, mother’s lack of education, violence, lack of social support, and restricted access to public policies regarding health and education). In contrast, the internal risk factors are associated with parental practices and parent-child interaction styles (e.g., abuse, neglect, negative communication, inconsistent discipline) [11]. In homeless families, many of those internal and external risk factors are simultaneously present, which accounts for their greater vulnerability [12]. From a nursing care perspective, vulnerability has always been considered a priority. As such, these families are a focus of interest for the nursing practice, often being the target of the aforementioned care [13].

A homeless person is an individual who, “regardless of his/her nationality, racial or ethnic origin, religion, age, sex, sexual orientation, socio-economic status, and physical and mental health status, is: homeless, living in a public space, in an emergency shelter, or in a precarious location; or homeless, living in a temporary accommodation destined for that purpose” (own translation, from: Ministério do Trabalho, Solidariedade e Segurança Social) [14] (p. 3925). 

The reasons for the families’ homelessness are different from those that lead to homelessness in adults. Most families are left homeless by: domestic violence [15,16,17]; repeated episodes of trauma and violence throughout life; child abuse; relationship breakups [15,16]; neighbourhood harassment; mental health problems involving parents and/or children [15,16,18]; special educational needs [15,16]; lack of family and social support networks; lack of access to statutory services [15,16,17,19]; social and emotional problems [17,20]; chronic instability; compromised health [19]; family head with a physical disability, with chronic health problems or with mental health problems [17]; poor financial situations and/or low income streams [21]; precarious housing situations; financial problems due to underemployment or unemployment; past mistakes (e.g., high school or college dropouts); and premature responsibilities, such as assuming a parental role at a young age [19]. The characteristics of these homeless families are heterogeneous. The most common are single-parent families led by (single) women [16,17,22]; with two children, generally under 11 years of age [16].

The lack of housing brings high social costs [23] and in families, has profound consequences, including the risk of health deterioration in family members [24,25]; the disruption of family dynamics, and the separation of parents and children [25]. Homeless families present multiple problems and needs that are interrelated (e.g., social, educational and health care) [15]. Homelessness is rarely an isolated event. Children from homeless families are prone to have a history of: low birth weight; anaemia; tooth decay; delayed immunizations; growth deficits; increased accident frequency, mainly injuries and burns [16]; developmental deficits; mental health problems [16,24,26,27]; behaviour disorders (e.g., sleep disorders, eating disorders, aggression and hyperactivity in young children); anxiety and post-traumatic stress [16]. A combination of multiple adverse childhood experiences may weaken their resilience [28] and they may become homeless as adults [28,29].

The protective factors for children belonging to homeless families include: school, due to its social stability, its routines and the parents’ sense of achievement when they are capable of maintaining their child in the same school [16]; and the positive relationships between parents and children, which can mitigate the negative effects of homelessness in childhood, by helping the children to better manage their emotions, by promoting self-regulation, by forming positive relationships and by improving the children’s functioning [30].

Homeless families face unique conditions that can affect the health and well-being of both the parents and the children. They may also affect parental practices, a better understanding of the homeless parents’ life experiences being necessary in order to increasingly meet the real needs of this population. The Canadian Homeless Press Observatory argues that the support given to families, through the strengthening of relationships, is essential to avoid situations of homeless youth [29].

The European Federation of National Organisations Working with the Homeless (*Fédération Européenned’ Associations Nationales Travaillant avec les Sans-Abri*—FEANTSA) suggests future research, highlighting the importance of studies in the following areas: homeless families, in order to further unveil the existing challenges and the strategies to be implemented, concerning the relationship with these families; parenting, in order to identify the constraints and strategies that homeless women (and men) face in their parental roles, as well as in the construction of their parental identities [31].

A preliminary search in the PubMed and CINAHL databases revealed the inexistence of systematic reviews on this topic and that none is currently in progress. Hence the necessity of aggregating knowledge to better comprehend the parental experiences of homeless individuals, and to identify useful factors for the monitoring of these vulnerable families. This will contribute to the understanding of how professionals can support families in managing their situation. 

It is important to highlight the importance of this work for the scientific area of nursing, since nurses are considered the health professionals closest to the population, playing a fundamental role in health promotion and disease prevention, especially among the vulnerable population, as is the case of homeless families. This work is part of the Special Issue of the International Journal of Environmental Research and Public Health, since it emphasizes the importance of nursing for society as promoters of the population’s health.

In this sense, we decided to conduct this systematic review, in order to better recognize the main research lines that have been explored up to this date, and to respond to the challenge proposed by the FEANTSA. For that purpose, we employed a scientific methodology which follows the Joanna Briggs Institute (JBI) protocol for systematic reviews of qualitative evidence [32].

The objective of this systematic review was to identify available qualitative evidence and to develop a framework to address the life experiences of homeless families in parenthood. We intended to answer the following research question, formulated according to the PICo acronym, [32] (Population, phenomenon of Interest and Context): What are the parenthood (I) experiences of the homeless (C) family (P)?

## 2. Methods

This systematic review adopts the method suggested by Moher, Liberati, Tetzlaff, Altman and the PRISMA group [33].

### 2.1. Inclusion Criteria

***Population—***This review considered all the studies that focus on homeless families constituted by adult parents (over 18 years of age) and their children (under 18 years of age). The parents and/or family correspond to the individuals who are responsible for providing care to the child/adolescent [32]. The following types of family were included: nuclear family (a man and a woman, who may or may not be legally married, with one or more biological/adopted children); reconstructed family (a couple in which at least one of the elements had a previous marital relationship and has a child who resulted from that relationship); single-parent family (a single parent and one or more children, being identified the gender of the person who represents the parental figure); extended family (a nuclear family, plus other relatives, or people who share with them bonds other than kinship) [10].

***Phenomenon of interest***—This review considered all the studies that address the parenthood experiences of homeless families, with parenthood being based on the evaluative categories of the parental role proposed by Figueiredo [10], and also the five parental functions of Cruz [9].

***Context***—This review considered studies conducted with homeless families. To that end, we employed the definition of “Homeless” recommended by the ENIPSSA [14].

***Type of studies***—Primary studies, of mixed or qualitative nature, published in Portuguese, French, English and Spanish, without a previously defined time span.

### 2.2. Exclusion Criteria

We excluded text (review) articles, opinion articles, letters to the editor, response letters, dissertations and reports, since priority was given to the numerous original research articles.

### 2.3. Search Strategy

The research strategy that was used aimed to find published studies indexed in certain databases. It comprised three stages. The first stage included an initial search in the PubMed and CINAHL databases, to analyse the keywords; it also encompassed the analysis of the MeSH Health Sciences descriptors in the respective query platform, as well as in CINAHL. The MeSH terms were then used in the PubMed database, while the corresponding MeSH headings were used in the CINAHL database. After selecting the relevant Boolean operators, descriptors, and indexed terms used to describe them, a second search was conducted in all the included databases—PubMed and CINAHL Complete (through EBSCOhost). The keywords which were used corresponded to the ones defined according to the PICo acronym (**P**opulation, phenomenon of **I**nterest and **C**ontext). The research strategy was sensitive to the specific characteristics of each selected database (Table 1).

The searches were carried out between January 9th and January 30th, 2019. Duplicates were removed and, subsequently, two independent reviewers performed an analysis of the terms and descriptors found in the titles and abstracts of the articles. Studies that did not meet the inclusion criteria were excluded. Afterwards, the selected articles were read in full. Those that did not satisfy the inclusion criteria, or corresponded to the exclusion criteria, were eliminated. Finally, in a third stage, the selected articles’ bibliographic references were analysed, in order to identify additional relevant studies. This review was not recorded in any systematic review database.

Before their integration in the final sample, all the selected articles were subjected to a methodological quality assessment.

### 2.4. Evaluation of the Methodological Quality

The methodological quality evaluation was conducted by two independent reviewers, using the JBI Critical Appraisal Checklist for Qualitative Research [32]. Disagreements between the reviewers were resolved through debating. The evaluation’s purpose was to analyse the strengths and weaknesses of each selected work. The reviewers critically appraised the articles, which were accepted for inclusion if there was congruence between the research methodology, the research’s objectives, and the methods used for data collection, representation and analysis.

### 2.5. Data Extraction

Data was extracted from all the works included in the review, using the JBI-QARI standardized data extraction tool [32]. The extracted data encompassed specific details about each research’s objectives, scientific area, type of study (design and data collection method), population, context, phenomena of interest, and significant findings for our research question and objective.

### 2.6. Data Synthesis

The qualitative research’s results were grouped and categorized based on the similarity of meaning, with at least two findings per category, using the JBI-QARI tool [32]. The different categories were created through a process of consensus between the reviewers. Data was then aggregated and subjected to synthesis, resulting in a single comprehensive set of findings that can provide a basis for evidence-based practice. The findings were classified as “credible” or “unquestionable” (according to their credibility level), using the JBI-QARI tool.

## 3. Results

We obtained a preliminary total of 358 articles through the database search. After duplicate removal, 349 works were subjected to relevance assessment based on their title and abstract. Of these, 248 were excluded after title evaluation and 64 were excluded after abstract evaluation, because they did not meet the established inclusion criteria. The remaining 37 studies were subjected to an eligibility assessment, and 26 were rejected, since they did not meet the inclusion criteria. Thus, a total of 11 articles were subjected to critical evaluation, with none being excluded. Figure 1 shows a PRISMA flowchart of the studies’ identification and selection process for inclusion in the review.

### 3.1. Methodological Quality 

All the 11 selected works were included in the analysis. These studies met the inclusion criteria and were considered methodologically rigorous, regarding the evaluation criteria. They offered useful descriptions of the daily life of homeless families. They also showed congruence between the research methodology, the research’s objectives, the methods used for data collection, representation and analysis, and the methods employed to interpret the results. Most of the data represented the participants’ opinions adequately. Despite the high quality of the selected articles, there was a lack of information in some of them. For instance, only 18% presented a statement contextualising the researchers’ cultural and/or theoretical background. Furthermore, only 54.55% of the reported conclusions were derived from data interpretation. Prior to the critical evaluation, the reviewers decided that these shortcomings were not essential criteria that needed to be met for the article’s inclusion. Table 2 presents the final assessment of the included works, specifying the research design’s accuracy, as well as the quality of the resulting reports.

Q1. Is there congruence between the declared philosophical perspective and the research methodology? Q2. Is there consistency between the research methodology and the research’s question(s)/objective(s)? Q3. Is there consistency between the research methodology and the data collection methods? Q4. Is there congruence between the research methodology and the data’s representation and analysis? Q5. Is there congruence between the research methodology and the results’ interpretation? Q6. Is there a statement that contextualises the researcher from a cultural or theoretical point of view? Q7. Are the researcher’s influence in the research and the inverse phenomenon addressed? Q8. Are the participants and their opinions adequately represented? Q9. Is the research considered ethical, according to the current criteria, or, for recent studies, is there evidence of ethical approval by an appropriate committee? Q10. Do the conclusions drawn in the research report derive from the data’s analysis/interpretation? Note: Q1, Q2, Q3, etc., refer to the questions included in the JBI Critical Appraisal Checklist for Qualitative Research [32].

Regarding the CONQual results, most of the selected studies obtained a moderate score [37,38,39,40,42], four reached a high score [15,25,34,36], one obtained a low score [35], and one achieved a very low score [41]. Despite their low classification, these last two articles were not eliminated, because their findings were unique and contributed to the knowledge of the phenomenon. Additional details can be seen in Appendix A, which are available in electronic format.

### 3.2. Results of the Extracted Data

Concerning the research design, the studies are mostly qualitative in nature [15,25,34,35,38,39,40,42]. Only three works present a mixed nature, with qualitative and quantitative approaches [36,37,41]. Two of the articles are grounded in theory [38,39], while the remaining did not declare any specific methodology, merely stating that they used a qualitative approach. The methods applied in the 11 articles were diverse, namely: qualitative individual interviews, family interviews, focus groups with parents, participant observation, diaries and reflection activities.

The works are scattered over time (from 1997 to 2018). After the first article, which appears in 1997, there is a large gap, whilst the remaining articles were published between the time period 2003 to 2018. The first article that appears after this gap is the nursing study. They were mainly conducted in the United States, except for two studies that were performed in the United Kingdom [15,41] and one study that was carried out in Australia [40]. The context of the studies corresponded to temporary/transitional shelters for nuclear families [15,25,34,36,37,41] or single-parent families led by homeless women [35,38,42]. Only one study was conducted in multiple contexts (shelters, motels and transition houses) for homeless nuclear families [39]. Another research was carried out with single-parent families led by homeless men, who were recruited for the study by agencies that provide support to the homeless [40].

As for the number of families interviewed, the majority were single-parent families led by women, with a total of 506 families, and in two studies the number of interviews was not mentioned [35,38]. Next, the largest number of records were single-parent families led by men with a total of 51 interviews, 40 of whom were part of the same study [40]. Finally, there were 22 nuclear families (couple with children) in two of the studies [15,41], and five interviews that were represented by relatives in one of the studies [39].The most frequent research field was mental health [15,25,35,36,37,41], followed by social sciences [34,38,40], occupational therapy [42], education (in one of the psychology studies) [35], and nursing [39]. Further details about the included studies are presented in Appendix A that will be available in electronic form.

### 3.3. Synthesis of the Qualitative Research’s Results

Taking into account the objective of the study, which sought to develop a structure to address the experiences of homeless families in parenthood, we extracted a total of 74 results from the 11 studies. Those findings were combined to form 12 categories based on similarity of meaning. The categories were grouped into two dimensions: “Parenthood Mediating Dimensions” and “Parenthood Supporting Dimensions”. As mediating dimensions, we considered situations of transition to a higher parenthood stage, where the transition’s inhibiting factors are highlighted. As supporting dimensions, we considered situations which reveal the transition process, emphasising the facilitating resources for that transition. The mediating dimensions include the categories “1—Insecurity”, “2—Lack of Privacy”, “3—Isolation”, “4—Stigma”, and “5—Disempowerment”, which support the categories “6—Education Difficulties”, “7—Changes in Parents’ Behaviour”, and “8—Changes in Children’s Behaviour”. On the other hand, the supporting dimensions include the categories “9—Context as a Facilitator”, “10—Relationship with Others”, and “11—Parents’ and Children’s Self”, which lead to the additional category “12—Parenting Motivation and Strategies” that addresses parenting issues.

Next, we will present the synthesis of our findings, in the form of a narrative text, followed by an overview of the obtained results associated with the aforesaid findings and the established categories.

### 3.4. Synthesis of the Findings

#### 3.4.1. Parenthood Mediating Dimensions

This synthesis derives from 53 findings and the following eight categories: “Insecurity”, “Lack of Privacy”, “Isolation”, “Stigma”, “Disempowerment”, “Education Difficulties”, “Changes in Parents’ Behaviour”, and “Changes in Children’s Behaviour”.

The mediating dimension of the “Insecurity” category is sustained by the following findings: lack of security; lack of hygiene; racism; intimidation; attacks from shelter residents; and negative influences from others. In the “Lack of Privacy” category, it is supported by: space limitations; restrictive environment; and the need for privacy to play. In the “Isolation” category, it is sustained by: shelters located far from family, school and city centre; separation from friends and family; distancing from children; and the existence of loneliness feelings. In the “Stigma” category, it is supported by: feelings of stigmatization, humiliation, disrespect, and judgment. The “Disempowerment” category is divided into 3 subcategories: “Routine Change”, sustained by the changes in the shelter’s routines, the difficulty in maintaining routines, and the involvement of other adults; “**Rule Enforcement**”, supported by shelter rules not consistent with previous parenthood options, as well as by the imposition of external rules; and “Parents’ Needs”, sustained by the need for practical, material and financial help, employment, education, transport, housing assistance, mental health promotion, pregnancy support, and time for oneself and one’s children, as well as by the inability to play the role of provider and guide. The “Changes in Children’s Behaviour” category includes: mental health problems; disobedience; mood swings; disrespect for authority; aggression; feelings of anguish, boredom and confusion; eating disorders; development regressions; concerns; and hiding the reality of living in shelters. The “Changes in Parents’ Behaviour” category encompasses the lack of availability and conditions to meet the child’s needs, as well as the dependence on the shelter’s resources. Finally, the “Education Difficulties” category includes: difficulty in establishing discipline; the need for help when dealing with school; difficulty in integrating the school’s community; difficulty in identifying school absenteeism as a problem; inability to help the child with school tasks, to maintain the child in a single school system throughout the year, and to identify academic problems; abdication of the responsibilities concerning school; absence of regular school attendance; and education barriers related to communication.

#### 3.4.2. Parenthood Supporting Dimensions

This synthesis derived from 21 findings and 4 categories: “Context as a Facilitator”, “Relationship with Others”, “Parents’ and Children’s Self”, and “Parenting Motivation and Strategies”. It demonstrates that the homeless situation experienced in temporary/transitional shelters seems to enhance parenthood issues in some families, since they lead to motivation and acquisition of strategies by the parents, in order to deal with those issues. Some parents perceive this “Context as a Facilitator”, because: it gives them access to services and resources; it is a safe place; it strengthens family practices and cohesion; and it is a stabilising experience that highlights the parental role. Additionally, they consider that it promotes the “Relationship with Others”, defending that: it is a social opportunity; it provides a sense of community and of collective parenting; it provides supportive relationships, positive encounters with community service providers, and social support from family and friends. In addition, it seems to encourage the “Children’s Self”, since a young person demonstrated having resilience and perseverance, despite being homeless. It also seems to promote the “Parents’ Self”, since they reveal: the ability to help their children to deal with emotions; the ability to identify academic problems; optimism; hope for the future; perseverance; self-efficacy; resilience; resistance; the ability to set goals; and that they can do without help. Despite the homelessness situation, in some families parenthood seems to be enhanced, and it is even verified that the children function as a source of “Motivation”, leading parents to seek improvement regarding their life situation and employment status, as well as to seek help to promote their mental health and maintain their morale. Parenthood also appears to feed the desire to move forward, in order to provide a better life for the respective children. Furthermore, to cope with the various challenges, the parents seek “Strategies”, such as: reading; writing a diary; keeping focus; attending church; and talking to the respective support team, friends and family.

Figure 2 shows the relationship between the categories and the synthesised findings, to illustrate the general synthesis process.

## 4. Discussion

This systematic review sought to gather the best available evidence on the parenthood experiences of homeless families. As stated previously, the aggregation of 11 qualitative research articles resulted in the finding of two types of parenthood dimensions. On the one hand, we found “Parenthood Mediating Dimensions” (corresponding to transitional states), which include: “Insecurity”, “Lack of Privacy”, “Isolation”, “Stigma”, and “Disempowerment” (the latter pertaining to routine changes, rule enforcement and the multiple needs of the parents). On the other hand, we found “Parenthood Supporting Dimensions”, which encompass: “Context as a Facilitator” (of access to services and resources, by acting as a stabilising experience for the family, and by highlighting the parental role), “Relationship with Others”, and finally the “Parents’ and Children’s Self”. In their turn, these categories led to relevant findings. The “Parenthood Mediating Dimensions” sustain changes in the parents’ and children’s behaviour, but, above all, they are associated with difficulties in education. Conversely, the “Parenthood Supporting Dimensions” lead to motivation and to the acquisition of strategies by the parents, with respect to parenting issues.

We found that some of the considered dimensions fit into the concept of parenthood recommended by the International Council of Nurses [1], as well as into the one suggested by Figueiredo [10]. Nevertheless, most of the dimensions corresponded to experiences, or feelings, reported by the parents. Some reflected difficulties in exercising the parental role, while others revealed protective factors related to the homelessness situation. This was one of the limitations we encountered in this study. In the “Parenthood Supporting Dimensions”, the transition process is revealed through: the need of establishing relationships with others, as sources of support and information; the interaction with others; and the use of resources and coping strategies, to deal with the situation. This is in line with the findings presented by Meleis [3].

We encountered another limitation in this study, related to the context of the included works (all were conducted in temporary or transitional shelters). The research should be extended to other contexts of homelessness, such as: the street/public spaces, precarious locations, pensions, and rooms paid for by the social services.

We also found that the families identified in the study fit the characteristics of the different types of homeless families, such as single-parent families led by (single) women [16,17]. Additionally, we identified multiple problems and needs in the studied families, which is in accordance with the results obtained by Anderson et al. [15], as well as multiple risk/protection factors (both internal and external), in line with the findings of Macana and Comim [11].

Regarding the limitations of this review, we stress that only articles published in English, Portuguese, French and Spanish were included, and there might have been relevant findings in works published in other languages. Another limitation relates to the use of a small number of databases, since its extension would have led to a broader research, and to the possibility of including other significant contributions. Another possible limitation concerns the priority given to the search for articles, thus excluding other kinds of text, such as opinion documents, dissertations and reports. Since the literature available for inclusion was limited to published original research articles, there is always the risk that relevant studies may not have been identified.

Despite the effort to find all relevant articles, some significant research works may have been overlooked. There is the possibility that both reviewers may have neglected important studies, during the screening process of the 358 articles initially obtained. On the other hand, the bibliographic search allowed the identification of articles produced in various professional fields, such as psychiatry, psychology, social sciences, occupational therapy, and nursing. This provided a broader picture of the homeless parents’ experiences.

Some of the obtained articles corresponded to secondary findings related to the parenthood experiences of homeless families, being thus included in the review. For instance, the purpose of one of the selected articles was to identify the parents’ and the employees’ perceptions, regarding their experience with the family support team, and also to recognise ways of improving this service in the future [15]. Although the focus was on the experience with the family support team, some of this study’s findings were considered relevant to the objective of the present systematic review, being, therefore, included in it. This would not have been possible using a more restrictive bibliographic search strategy.

When we critically evaluated the 11 included articles, they revealed a variety of strengths and limitations. All works showed coherence between the research methodology and the research’s question(s)/objective(s), the data collection methods and the results’ interpretation. Nonetheless, two of the studies did not present congruence between the stated philosophical perspective and the research methodology [35,39], and there were two articles that did not present congruence between the research methodology and the data’s representation and analysis [35,41].

The absence of an appropriate statement enhanced the complexity of evaluating the congruence between the philosophical perspective and the research methodology. Additionally, the inexistence of congruence, regarding the data’s presentation and analysis, increased the difficulty of the data’s evaluation. Although the reviewers did not consider these phenomena a reason for exclusion, the congruence between the philosophical perspective and the research methodology, as well as between the research methodology and the data’s representation and analysis, would have been preferable, to strengthen the studies’ credibility. 

Data were collected through participant observation, individual interviews, written diaries, reflection activities, and group interviews/focus groups. The fathers’ perspectives were under-represented, because most of the interviewees belonged to single-parent families led by women, thus highlighting the perspective of homeless mothers. However, in one of the studies, it was possible to analyse exclusively the male perspective, since it encompassed single-parent families led by men [40]. This unequal representation might be considered a limitation, because homeless fathers may have different experiences from homeless mothers. Even so, it may provide a realistic picture of homeless families.

Most of the studies were conducted in the United States, except for two which were carried out in the United Kingdom [15,41], and one that was performed in Australia [39]. This geographical distribution can result in limitations related to cultural differences, concerning the way homeless parenthood is viewed. It is important to note that these results can also refer to refugee families, which are becoming more important today. Such divergences may be relevant, since they can have implications for the practice in various settings and cultures. Nonetheless, we consider the studies’ heterogeneity a strength, because it represents the complexity and diversity of the parenthood experiences in homelessness situations.

## 5. Conclusions

This review’s findings contributed to the scientific knowledge about the complexities of homeless families/fathers/mothers and the challenges faced by them. Homeless parents living in temporary or transitional shelters are confronted with the intricacy of parental experiences influenced by insecurity, lack of privacy, isolation, stigma and disempowerment. On the other hand, to some families, the homelessness experience lived in temporary or transitional shelters seems to enhance parenthood, since they regard that context as a facilitator of the parental role, which promotes family cohesion, the relationships within the family and with others, as well as the parents’ and the children’s self. During the complicated journey of homeless parenthood, the parents use coping strategies and see their children as a great source of motivation, where they find strength to fight against the multiple difficulties they experience.

This review adds scientific knowledge on the experiences of homeless families, by revealing several parenthood mediating and parenthood supporting dimensions. However, the specific knowledge about parental experiences is still limited. The concept of homeless parenthood is scarcely researched; there exists a need for further studies in other homeless contexts, and in the field of Nursing, in order to adapt nursing responses to those families. While conducting this review, we found a reduced number of articles about the studied topic, which justifies the presented need and encourages additional research in the area of homeless parenthood. We stress the need for further research on paediatric nursing care, in the above-mentioned area. Regarding our methodological approach, we conclude that it was appropriate, since it allowed answering the research question.

## Figures and Tables

**Figure 1 ijerph-17-02712-f001:**
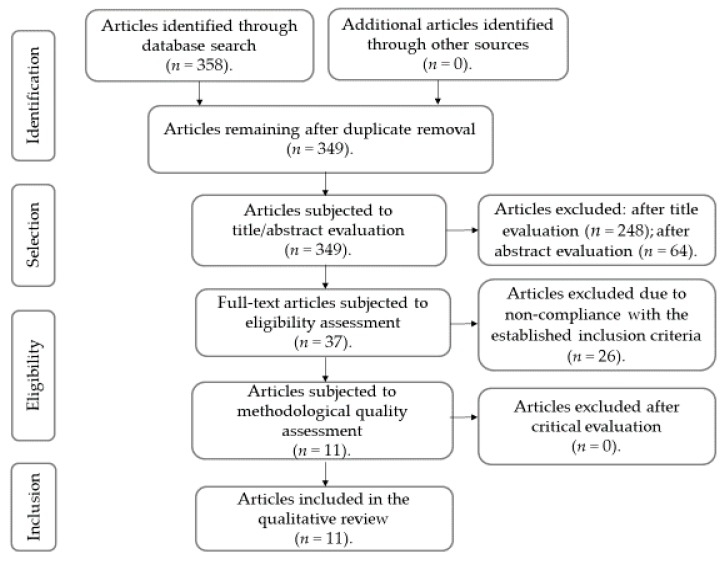
PRISMA flowchart detailing the search’s results and the studies’ selection/inclusion process (based on the method suggested by Moher, Liberati, Tetzlaff, Altman and the PRISMA group) [33].

**Figure 2 ijerph-17-02712-f002:**
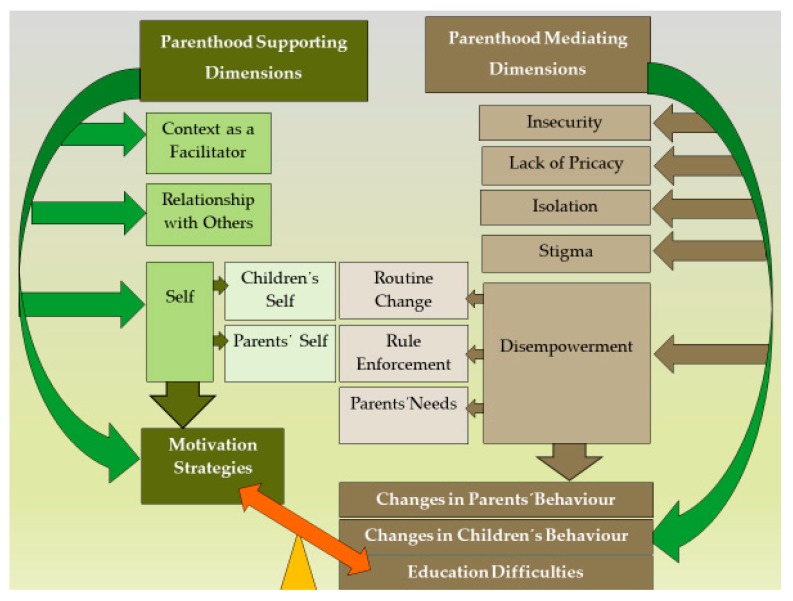
Overview of the synthesised findings.

**Table 1 ijerph-17-02712-t001:** Search strategy.

Database	Search Strategy
PubMed	(((((((((((((((((parenting) OR parenting styles) OR parenting practices) OR parent education) OR childrearing) OR family practice) OR parenting education) OR parenting education) OR parental behaviour) OR parent* behaviour) OR parent* education) OR parent* attitudes) OR altered parenting) OR parent* roll) OR parenthood)) AND ((((homeless persons) OR homeless people) OR homelessness) OR vulnerable populations)) AND (((((((((((((parent* OR biological parents) OR child of impaired parents) OR marriage) OR spouses) OR mothers) OR fathers) OR parents) OR only child) OR nuclear family) OR single parent) OR single parent family) OR family)
CINAHL	(parent* OR biological parents OR child of impaired parents OR marriage OR spouses OR mothers OR fathers OR parents OR only child OR nuclear family OR single parent OR single parent family OR family) AND (homeless persons OR homeless people OR homelessness OR vulnerable populations) AND ((MH “Parenting”) OR “parenting” OR (MH “Altered Parenting (NANDA)”) OR (MH “Parenting (Iowa NOC)”) OR (MH “Parenting Alteration (Saba CCC)”) OR (MH “Parenting Education”) OR (MH “Risk for Altered Parenting (NANDA)” OR parenting styles OR parenting practices OR child-rearing OR family practice OR parental behaviour OR parent* behaviour OR parent* education OR parent* attitudes OR parent* role OR parenthood)

**Table 2 ijerph-17-02712-t002:** Final evaluation of the included studies (according to the method used by Lockwood) [32].

Quote	Q1	Q2	Q3	Q4	Q5	Q6	Q7	Q8	Q9	Q10
Anderson et al. 2006 [15]	Y	Y	Y	Y	Y	N	Y	Y	N	Y
Anthony et al., 2017 [34]	Y	Y	Y	Y	Y	N	Y	Y	Y	Y
Cosgrove and Flynn, 2005 [35]	N	Y	Y	NC	Y	N	Y	Y	N	NC
Haskett et al., 2018 [36]	Y	Y	Y	Y	Y	Y	Y	Y	Y	Y
Holtrop et al., 2015 [37]	Y	Y	Y	Y	Y	N	Y	Y	Y	NC
Lindsey, 1997 [38]	Y	Y	Y	Y	Y	N	N	Y	N	NC
Morris and Butt, 2003 [39]	N	Y	Y	Y	Y	N	Y	Y	Y	Y
Roche et al., 2018 [40]	Y	Y	Y	Y	Y	N	Y	Y	Y	Y
Sylvestre et al., 2018 [25]	Y	Y	Y	Y	Y	Y	Y	Y	Y	Y
Tischler et al., 2004 [41]	Y	Y	Y	NC	Y	N	Y	NC	Y	N
Vanleit et al., 2006 [42]	Y	Y	Y	Y	Y	N	Y	Y	Y	N
% Yes	81.82	100	100	81.82	100	18.18	90.91	90.91	72.73	54.55

N: No; NC: Not Conclusive; Y: Yes.

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
