# Peer review of "Experiences of Homeless Families in Parenthood: A Systematic Review and Synthesis of Qualitative Evidence"

_ijerph, 2020, doi:10.3390/ijerph17082712_

Round 1

Reviewer 1 Report

I have reviewed the paper:  Experiences of Homeless Families in Parenthood: A Systematic Review of Qualitative Evidence

Manuscript ID: ijerph-753018

This study provides content of interest to nursing and society.

However, the paper has a number of important limitations that I believe can be improved by the authors.

In general, the paper should improve:

1-The justification in the special issue

2-Written English

3-The format, typeface of various paragraphs, tables, figures etc.

4-The format and quality of the references.

5-The paper is not in the IJERPH template. I can't see the lines. For this reason, I attach the comments in the pdf as well. I ask the authors to attach the IJERPH template for the next revision and underline the changes in the text in yellow.

On the other hand, its content is of great interest to the scientific community. I believe that if the authors incorporate all the review aspects, the paper will gain in quality.

Key words

I suggest to the authors, that a key word should be nursing

Abstract

The abstract should be reoriented. the importance of nursing in this matter must be clearly justified.

The impact of this paper must be clearly justified in the special issue of Special Issue “Nursing and Society”

https://www.mdpi.com/journal/ijerph/special_issues/nursing_society

On the other hand, the justification for this topic, the lack of studies and its impact must be clearly reflected.

Introduction

Authors should review English, and punctuation

references are missing

the references in the text are not correctly numbered according to IJERPH standards

The introduction should add more references in English.

Authors should thoroughly review the reference format.

Authors should clearly point out the impact, that is, the added value of this article for the special issue of

“Nursing and Society”

https://www.mdpi.com/journal/ijerph/special_issues/nursing_society

Methods

Sentence too long

Authors should review the different types of letters, and the format of the table, since it does not comply with the IJERPH standards

Authors have not reflected years of systematic review

Results

In the results section, the authors must remember the object of the study.

Figure 1 is cut into 2 pages. Its typeface is different from that of the text.

Table 2 has different types of letters.

Review the IJERPH model to cite references

Table 3 has different fonts, and is cut into several pages.

Authors should improve the format of this table, which can be seen on one page. It is suggested horizontally.

The table has different fonts, and is cut into several pages.

Table 5 has different fonts, and is cut into several pages.

If different pages are necessary, the authors must put the first row of the table again.

They should look at the IJERPH format

Figure 2 needs to be greatly improved. It cannot be seen correctly.

Discussion

The discussion is correctly oriented.

Authors should review written English, and the reference model

6-Conclussions

The conclusion is correctly oriented.

References

References should be thoroughly reviewed by the authors.

Firstly, the format does not conform to IJERPH standards

Second, they should put as many DOI links as possible. Several references do not have the link or DOI. Others have a broken link.

Finally, there is a majority of Portuguese references. Authors are recommended to have more literature in the English language.

The typeface of the references does not conform to IJERPH standards, and is different from the rest of the text.

Authors should carefully review the entire format of the references. There are a large number of mistakes.

Reviewer 2 Report

The topic is unique and the presented results can be used (and cited) by other studies. The method of conducting the review is compatible with modern standards.

The two most serious comments relate to how the objective is formulated and to the scope of information presented in the introduction.
• The objective is too general, it does not reflect the scope of research. It should be changed or expanded. Instead of the phrase "to identify the available scientific evidence about the experiences of homeless families in parenthood" I suggest "to identify available qualitative evidence and to develop a framework to address experiences of homeless families in parenthood ". Changing the title to include “systematic review and synthesis of qualitative evidence” could also be considered.
• The review article may have an above-standard length, but in this case the introduction should be significantly shortened. I suggest e.g. only briefly refer to family functions.

Some changes in the summary should be introduced: redrafting the aim of the study (as above), providing the initial number of identified papers, describing the type of selected papers and also excluding the last sentence of the abstract.

Other comments relate to clarification or editing issues.

  • When defining the research question, reference was made to the PICO approach, which is known in EBM, but in a different sense (patient, intervention, comparison, and outcome). Please provide the appropriate reference.
  • This is not required, but please provide information on whether the review was registered in any database of systematic reviews.
  • The PRISMA methodology should be mentioned in the section on methods with appropriate citation.
  • It is worth adding that these results may also refer to refugee families, which is becoming more important nowadays. It may be included in the last paragraph, before the conclusions.
  • Figure 1 is incomplete in this version of pdf file.
  • Occasionally, a description of the methods appears in the results section (e.g. page 10) – please revise
  • Please clarify (or adjust) the order of papers in Table 4 (at present the papers are listed neither by alphabetical order nor by the year of publication).
  • It has not been clarified which country the results come from. This can be added in the third column of Table 5
  • Part 3.2 does not provide the number of families (subjects) in the studies presented
  • Due to the large volume of text, you may consider including some tables as supplementary material (e.g. Table 5) that would be available only in electronic form
  • References have a different font. Please bold the year of publication

Table 5 – leave only horizontal lines.

Round 2

Reviewer 1 Report

I carry out the second review of the paper:

Experiences of Homeless Families in Parenthood: A Systematic Review of Qualitative Evidence

Manuscript ID: ijerph-753018

The authors have improved the paper by incorporating several of the suggestions from the first review.

But there are still some suggestions that have not been incorporated, which I list below:

Introduction

Reference?

The reference that relates to the entire paragraph written below is not set

The authors have not corrected this suggestion, noted in the first review.

They should review the citation style of the references throughout the text.

This suggestion was noted in the first review

For example:

Sometimes the authors write (16-17)

and others (16.17)

It's an example. When the references are followed, the authors should consult the IJERPH norms and put a comma, or hyphen -

But always the same decision

Methods

Figure 1 should be revised. I point out in yellow

References

References should be carefully reviewed.

Authors have a high number of errors

I have pointed out only a few so far, but there are many incorrect references.

Some of the errors:

The doi don't work

Access dates indicate dates from years ago

Spaces, commas, etc., after or before the years

etc

Furthermore, a suggestion to the authors for following publications:

1-The papers must be sent in the journal template. Here the lines appear, and the reviewers can indicate in which line is the content that should be corrected.

2-In addition to underlining the corrected content, the authors must respond in one word to each of the reviewers' comments, indicating in which line they have incorporated the correction, and the text.

On this occasion, I have proceeded to review the content of the paper delighted with the investigation. This is only a suggestion for future papers.

Author Response

Dear Reviewer

Thank you very much for the revision that was made and for the suggestions given.

Concerning the given suggestions:

  • The spell check was made;
  • The reference to the whole written paragraph in the introduction on line 65 was revised;
  • The citation style throughout the text was revised (lines 121, 123 and 125) as well as the spaces and commas;
  • Figure 1 was revised;
  • The references were revised (doi, access dates, spaces and commas).

All the changes were highlighted in yellow in the text.

Kind regards